# The arginine methyltransferase PRMT5 promotes mucosal defense in the intestine

Juan E Hernandez[1], Cristina Llorente[2], Shengyun Ma[1], Kiana T Miyamoto[3], Saptarshi Sinha[1], Scarlet Steele[1], Zihui Xiao[1], Ching-Jung Lai[1], Elina I Zuniga[3], Pradipta Ghosh[1,2], Bernd Schnabl[2,4], Wendy Jia Men Huang[1]

**PRMT5 is a type II arginine methyltransferase abundantly expressed in the colonic epithelium. It is up-regulated in inflammatory bowel disease and colorectal cancer. However, its role in mucosal defense against enteric infection has not been studied. Here, we report that *Prmt5* in the murine colon is up-regulated in response to *Citrobacter rodentium* infection. Pathogen clearance in mice with haploinsufficient expression of *Prmt5* is significantly delayed compared with wildtype littermate controls. Transcriptomic analyses further reveal that PRMT5 regulates the expression of canonical crypt goblet cell genes involved in mucus production, assembly, and anti-microbial responses via methyltransferase activity–dependent and –independent mechanisms. Together, these findings uncover PRMT5 as a novel regulator of mucosal defense and a potential therapeutic target for treating intestinal diseases.**

## Introduction

The protein arginine methyltransferase (PRMT) family methylates arginine residues on target proteins, a common post-translational modification previously implicated in transcription, RNA processing, mRNA translation, signal transduction, and cell fate decisions (1, 2). In the intestine, PRMT5 is up-regulated in colorectal cancer and its expression positively correlates with a poorer prognosis (3). Cell culture studies suggest that PRMT5 contributes to colorectal cancer cell growth by transcriptionally activating oncogenes and/or repressing tumor suppressive protein functions (4, 5, 6, 7). However, the physiological role of PRMT5 in the colonic epithelium under homeostasis and during enteric infection has not been studied.

Goblet cells in the intestinal epithelium keep luminal commensal microbes in check and defend against food-borne pathogens (8). Alterations in their differentiation and function are often observed in inflammatory bowel disease (IBD) patients (9, 10) and

animal models of colitis (11, 12, 13, 14). Colonic goblet cells can be divided into multiple subpopulations based on gene expression. Although noncanonical goblet cells are characterized by genes involved in nutrient absorption and metabolite transport (15), canonical goblet cells are enriched with genes encoding the structural components of the mucus layer, including Mucin 2 (MUC2) and Fc-γ–binding protein (FCGBP), and enzymes that regulate the mucus structural arrangement, such as the chloride channel accessory 1 protein (CLCA1). Mucin-deficient mice develop spontaneous colitis (8, 16). When challenged with *Citrobacter rodentium*, a murine model of enterobacterial infection, mucin-deficient mice have a greater bacterial burden, higher intestinal epithelial barrier disruption, and an increase in mortality (17).

Goblet cells can also be subdivided based on their location within the intestinal epithelium. Goblet cells at the apical surface (intercrypt icGCs) secrete mucus that fills the spaces between crypt openings. Loss of icGCs in *Spdef*-deficient mice increases susceptibility to dextran sulfate sodium (DSS)-induced colitis (15). Crypt-residing goblet cells are responsible for secreting impenetrable mucus plumes that cover the crypt openings (15). However, the molecular regulators of the different goblet cell subsets and their functions remain largely unknown.

Here, we report that *Prmt5* is up-regulated in IBD and in response to *C. rodentium* infection in mice. Mice with reduced PRMT5 levels show a diminished ability to clear *C. rodentium* infection and sustain greater mucosal damage. Transcriptomic studies further reveal that PRMT5 regulates the expression of canonical crypt goblet cell genes involved in antimicrobial defense and mucus assembly. Mice with reduced PRMT5 levels have impaired colonic mucus production capacities and harbor aberrant inner mucus layer structure. Interestingly, although PRMT5 promotes the gene expression of the mucus assembly factor CLCA1 in an arginine methyltransferase activity–dependent manner, its regulation of the mucus structural proteins FCGBP and MUC2 is methyltransferase activity–independent. These results highlight the important role of PRMT5 in mucosal defense and provide novel targets for intestinal diseases.

[1]Department of Cellular and Molecular Medicine, University of California San Diego, La Jolla, CA, USA  [2]Department of Medicine, University of California San Diego, La Jolla, CA, USA  [3]Division of Biological Sciences, University of California San Diego, La Jolla, CA, USA  [4]Department of Medicine, Veterans Affairs San Diego Healthcare System, San Diego, CA, USA

Correspondence: wendyjmhuang@health.ucsd.edu

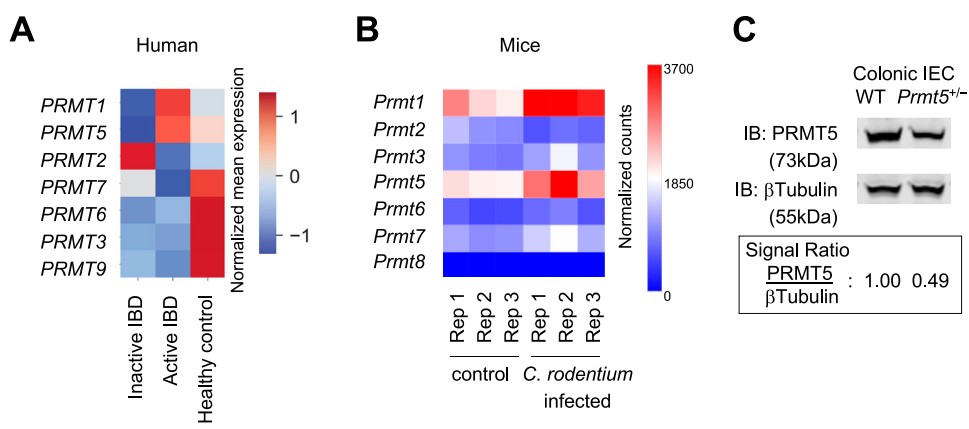

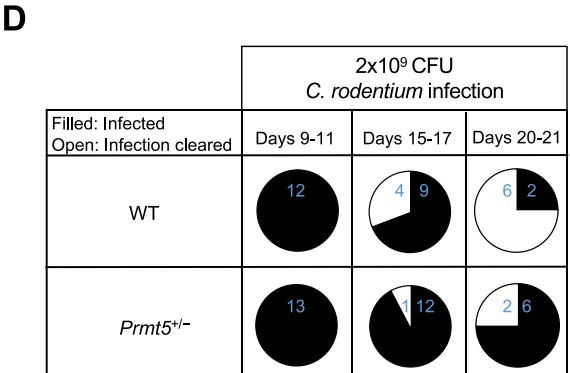

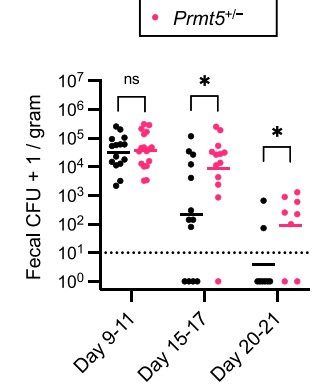

**Figure 1. PRMT5 promotes mucosal defense against *Citrobacter rodentium*.**
**(A)** Transcript abundance of genes encoding PRMT family members in colonic epithelium from active IBD, inactive IBD patients, and healthy controls (GSE179128). Scale bar indicates StepMiner-normalized transcript mean expression values. **(B)** Normalized RNA-seq read counts of PRMT family members in the colon of control and *C. rodentium*–infected mice (GSE100546). **(C)** Representative Western blot of PRMT5 and *β*-tubulin in colonic IEC whole cell lysates from WT and *Prmt5*⁺/⁻ mice. Experiments were repeated three times using independent biological samples with similar results. PRMT5 levels were calculated as PRMT5 signals over those of *β*-tubulin. **(D)** *C. rodentium* clearance status of WT and *Prmt5*⁺/⁻ mice between days 9 and 21 post-infection (filled: incomplete clearance; open: complete clearance; blue: number of mice represented). **(E)** Fecal *C. rodentium* CFU from mice on days 9–11, 15–17, and 20–21 post-infection. The dashed line indicates the limit of detection. Graph represents results from three independent infection experiments combined. The first experiment included WT (n = 4) and *Prmt5*⁺/⁻ (n = 5) mice. The second experiment included WT (n = 3) and *Prmt5*⁺/⁻ (n = 3) mice. The third experiment included WT (n = 6) and *Prmt5*⁺/⁻ (n = 5) mice. Data shown are geometric means. *P < 0.05, n.s., not significant (Mann–Whitney U test). **(F)** Left: representative H&E staining of colonic tissues from WT and *Prmt5*⁺/⁻ mice harvested on days 13 and 16 post-*C. rodentium* infection. Right: overall pathology scores of colonic sections from two independent infection experiments combined. The first experiment included WT (n = 3) and *Prmt5*⁺/⁻ (n = 4) mice. The second experiment included WT (n = 5) and *Prmt5*⁺/⁻ (n = 2) mice. Data shown are means ± SD. *P < 0.05 (t test). Scale bars represent 200 *μ*m.

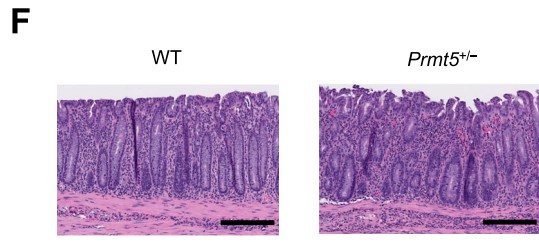

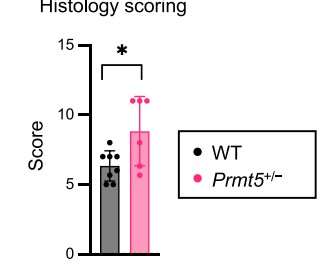

# Results

## PRMT5 is up-regulated in the inflamed colon

*PRMT5* is one of two members of the PRMT methyltransferase family up-regulated in the colonic epithelium from patients with active IBD compared with patients with inactive disease and healthy individuals (Figs 1A and S1). In the murine intestinal epithelium, *Prmt5* is one of the most highly expressed arginine methyltransferase family members under steady state (Fig S2A). Immunohistochemistry analysis revealed abundant PRMT5 protein in the nuclei of cells with morphologies corresponding to those in the epithelial secretory lineages, including the mucin vacuole–

containing goblet cells, tuft cells, and enteroendocrine cells (Fig S2B). In the murine model of *C. rodentium* infection-induced colitis, which shares the epithelial barrier dysfunction and inflammation evident in human IBD (18), *Prmt5* was similarly up-regulated (Fig 1B). These results suggest that PRMT5 may have a role in host defense against enteric infections and intestinal inflammation.

## Delayed *C. rodentium* clearance in *Prmt5*⁺/⁻ mice

To investigate the function of PRMT5 in vivo, we used the *Prmt5*⁺/⁻ mice, a heterozygote model, as the complete loss of *Prmt5* results in embryonic lethality (19). Gender-matched and cohoused *Prmt5*⁺/⁺ (WT) and *Prmt5*⁺/⁻ littermates showed comparable body weight (Fig S3A).

Western blot analysis of total cell lysates from isolated colonic IECs confirmed ~50% reduction in PRMT5 protein level in cells from *Prmt5*<sup>+/−</sup> mice (Fig 1C). To assess intestinal barrier function, 4 kD fluorescein isothiocyanate–dextran (FITC-dextran) were administered to WT and *Prmt5*<sup>+/−</sup> littermates by oral gavage. Four hours post administration, similar levels of FITC-dextran were found in the WT and *Prmt5*<sup>+/−</sup> bloodstream, suggesting that PRMT5 is likely dispensable for regulating protein-size macromolecule passage in the intestine (Fig S3B).

To determine the role(s) of PRMT5 in host defense against enteric infection, WT and *Prmt5*<sup>+/−</sup> littermates were challenged with a non-lethal dose of *C. rodentium*. At the peak of infection (days 9–11), WT and *Prmt5*<sup>+/−</sup> mice showed comparable levels of fecal *C.*

*rodentium* (Fig 1D and E). By 15–17 d post-infection, 31% of the WT mice had no detectable fecal *C. rodentium* compared with 8% among the *Prmt5*<sup>+/−</sup> mice (Fig 1D and E). By days 20–21, 25% of the *Prmt5*<sup>+/−</sup> mice achieved complete clearance compared with 75% in WT mice (Fig 1D and E). Overall, the fecal *C. rodentium* loads in WT mice that remained infected were lower than those found in *Prmt5*<sup>+/−</sup> mice. Colonic tissue sections harvested from *Prmt5*<sup>+/−</sup> mice showed higher pathology scores, consistent with impaired pathogen clearance (Fig 1F). Notably, the delay of *C. rodentium* clearance in the *Prmt5*<sup>+/−</sup> mice was not associated with changes to IFNγ, IL-17A, and IL-22 production capacities in the colonic lamina propria innate lymphoid cells (Fig S4). Taken together, these results indicate that PRMT5 supports host defense against enteropathogenic infection.

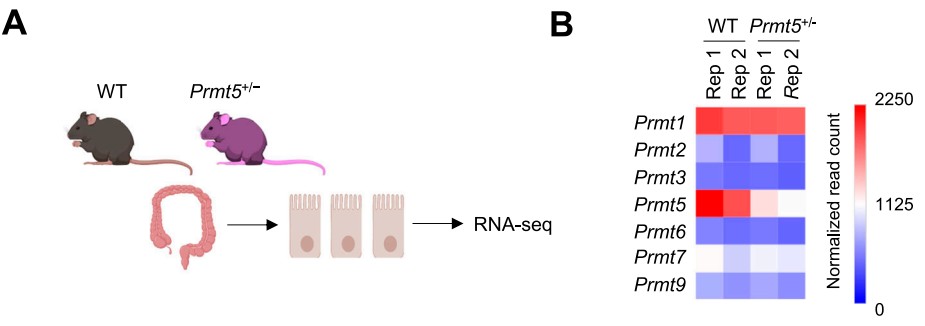

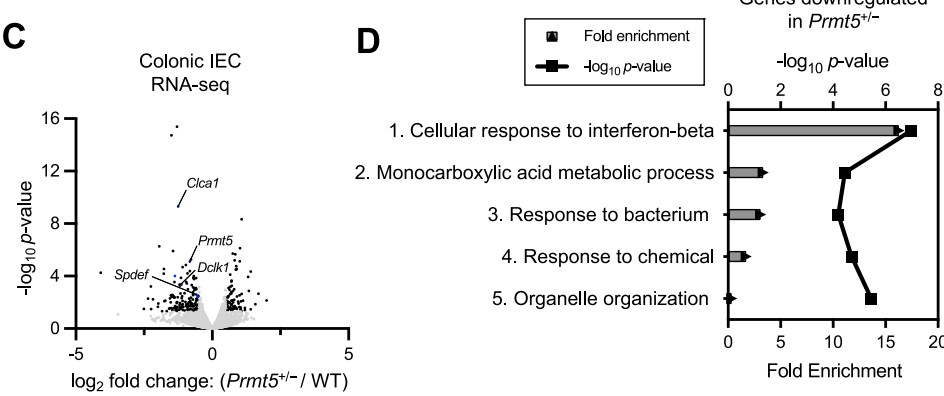

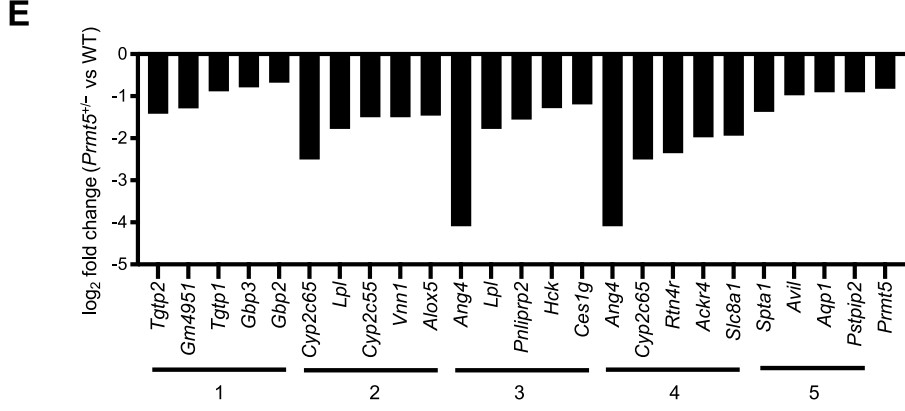

**Figure 2. PRMT5-dependent gene pathways in colonic epithelial cells.**
**(A)** Workflow of colonic epithelial cell isolation for global transcriptomic analysis. **(B)** Heatmap of normalized RNA-seq read counts of PRMT family members expressed in colonic IECs from two pairs of WT and *Prmt5*<sup>+/−</sup> cohoused littermates. **(C)** Differential gene expression in colonic IECs from two pairs of WT and *Prmt5*<sup>+/−</sup> cohoused littermates. Black dots: PRMT5-dependent transcripts defined as $\log_2$ fold changes ≥0.5 or ≤−0.5 and *P*-value < 0.05 as determined by DESeq2. **(D)** Top 5 gene ontology pathways enriched among genes downregulated in *Prmt5*<sup>+/−</sup> colonic IECs. **(D, E)** $\log_2$ fold changes of select PRMT5-regulated genes from each of the pathways from (D).

## PRMT5 regulates select colonic epithelial gene programs

Based on the abundant expression of PRMT5 in the IEC nucleus (Fig S2B), we hypothesized that PRMT5 may regulate transcription and/ or processing of colonic epithelial genes involved in mucosal defense to confer protection against enteropathogenic infection. To test this possibility, we performed bulk RNA-seq of colonic IECs from two pairs of gender-matched and cohoused WT and *Prmt5*$^{+/-}$ littermates under steady state (Fig 2A). We confirmed *Prmt5*$^{+/-}$ colonic IECs expressed significantly fewer *Prmt5* transcripts, and the expressions of the other PRMT arginine methyltransferase family members were not affected (Fig 2B). Differential gene expression analysis revealed 232 PRMT5-regulated colonic transcripts (Fig 2C and Table S1). Gene ontology pathway analysis of the 153 genes down-regulated in *Prmt5*$^{+/-}$ colonic IECs revealed enrichments for molecules involved in bacteria and IFN responses (Fig 2D and E). No significant gene ontology pathway was detected among the 79 genes that were up-regulated in *Prmt5*$^{+/-}$ colonic IECs.

Interestingly, goblet cell and tuft cell-enriched genes were the most significantly down-regulated in the *Prmt5*$^{+/-}$ colonic epithelium (Fig 3A and B). qRT-PCR analysis of additional samples from independent experiments confirmed reduced levels of transcripts encoding PRMT5, anti-microbial proteins ZG16 and ANG4, and the

tuft cell marker DCLK1 in the *Prmt5*$^{+/-}$ colon (Fig 3C). There was also a modest decrease in transcripts encoding the anti-microbial protein RELMβ (*Retnlb*), but no significant changes in the transcripts encoding the goblet cell master transcription factor KLF4 (20) or a marker of proliferation KI67. These results suggest that PRMT5 promotes goblet and tuft cell gene programs in the colonic epithelium. Interestingly, alterations in these goblet and tuft cell gene programs in the *Prmt5*$^{+/-}$ colon were not due to a reduction in goblet and tuft cell numbers. Periodic acid–Schiff (PAS) and DCLK1 staining of colon sections showed similar goblet cell numbers, tuft cell numbers, crypt densities, and crypt heights in WT and *Prmt5*$^{+/-}$ littermates (Figs 4A and S5). These results suggest that PRMT5 haploinsufficiency does not interfere with gross colonic epithelium organization, goblet cell, and tuft cell establishment under steady state.

## PRMT5 regulates mucus production and assembly

Colonic goblet cells can be divided into multiple subpopulations (15). Therefore, we next investigated whether PRMT5 regulates a common goblet cell gene program and/or subset-specific functions. GSEA revealed a significant dependency of the canonical goblet cell and crypt goblet cell–enriched genes on PRMT5 (Fig 4B and C). qRT-PCR analysis of additional samples from independent experiments

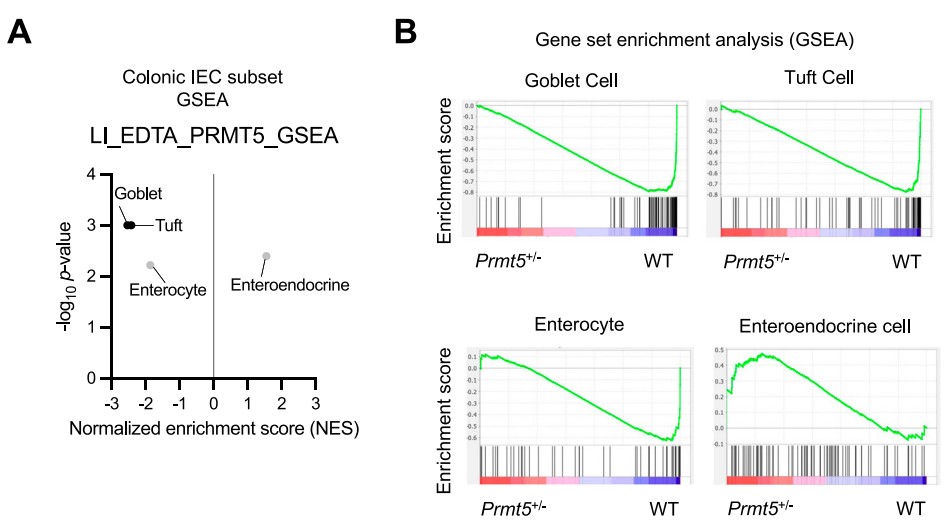

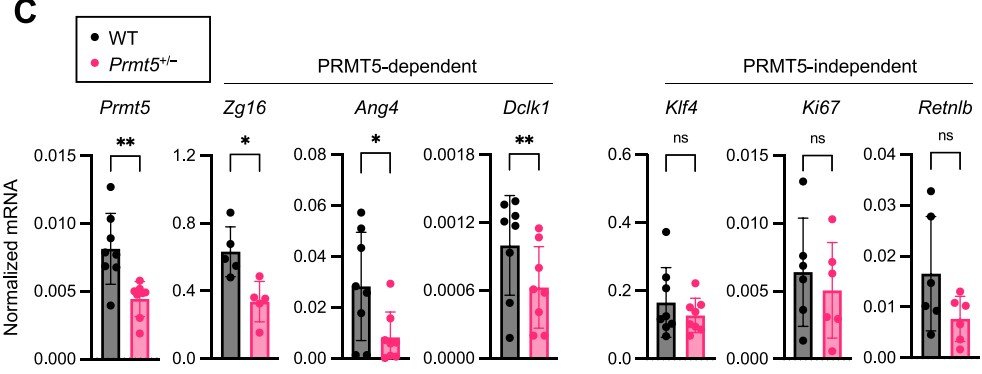

**Figure 3. PRMT5 regulates IEC subset-specific gene programs.**
**(A)** Plot summarizing the GSEA results on the indicated colonic epithelial cell gene subsets in WT and *Prmt5*$^{+/-}$ colonic IECs. NES, normalized enrichment score. **(B)** GSEA enrichment plots on the indicated colonic epithelial cell gene subsets in WT and *Prmt5*$^{+/-}$ colonic IECs. **(C)** Normalized mRNA expressions of select genes from independent pairs of WT (n = 5 or 8) and *Prmt5*$^{+/-}$ (n = 5 or 8) colonic IECs. Each dot represents the result from one mouse. Data shown are means ± SD. *$P$ < 0.05, **$P$ < 0.01, n.s., not significant ($t$ test).

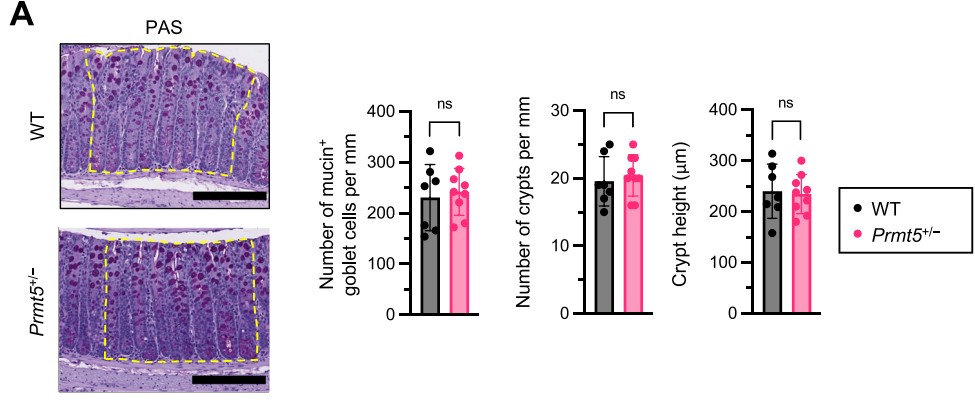

**Figure 4. PRMT5 promotes the expression of genes enriched in crypt-residing canonical goblet cells.**
**(A)** Left: representative WT and *Prmt5*⁺/⁻ colons stained with Periodic acid–Schiff staining. Boxes with yellow dotted borders represent regions used for quantification. Right: summarized colonic goblet cell (PAS^hi) counts, crypt densities, and crypt heights in WT (n = 7) and *Prmt5*⁺/⁻ (n = 9) mice. Data shown are means ± SD. n.s., not significant (*t* test). Scale bars represent 200 μm. **(B)** GSEA enrichment plots of canonical, noncanonical, crypt, and surface goblet cell gene sets in WT and *Prmt5*⁺/⁻ colonic IECs. NES, normalized enrichment score; NOM *P*, normalized *P*-value; FDR, false discovery rate. **(C)** Left: working model of PRMT5-regulating goblet cell subset–specific programs. Right: top 10 PRMT5-regulated crypt canonical goblet cell-enriched genes from B.

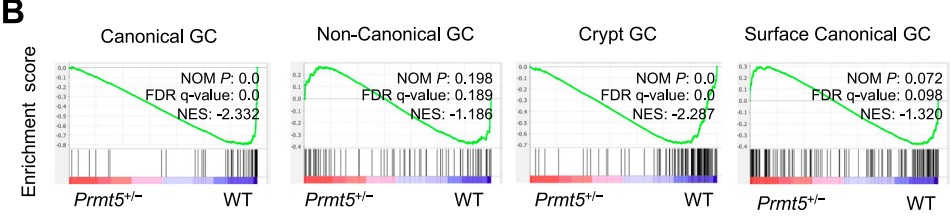

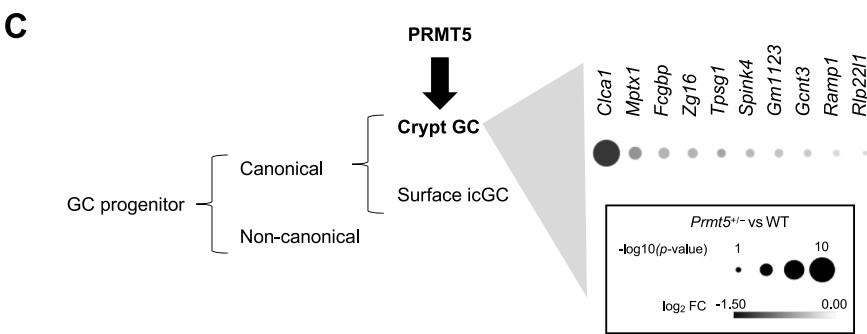

confirmed reduced levels of *Clca1*, *Fcgbp*, and *Muc2* in the *Prmt5*⁺/⁻ colon at steady state (Fig 5A). Western blot analysis of colonic epithelium whole cell lysates revealed significant reduction of FCGBP protein levels and slight reductions in MUC2 and CLCA1 protein levels in the *Prmt5*⁺/⁻ mice (Fig S6). As the anti-CLCA1 antibodies were also suitable for intracellular staining, we performed flow cytometry analysis and confirmed a significant reduction in the proportion of CLCA1-expressing colonic IECs in the *Prmt5*⁺/⁻ mice (Figs 5B and C and S7). Similar to steady state, *Clca1* transcripts were also significantly lowered in the *Prmt5*⁺/⁻ colonic epithelium post-*C. rodentium* infection (Fig S8). *Muc2* and *Fcgbp* transcripts were slightly reduced in the *C. rodentium*–infected *Prmt5*⁺/⁻ tissue (Fig S8). Similar to in vivo results, cultured MC38 murine colonic epithelial cells with higher PRMT5 protein levels also correlated with augmented CLCA1 abundance (Fig S9A and B). Notably, *Prmt5* expression in MC38 cells was not altered by serum amyloid protein (SAA1), IL-23, IL-1β, and/or heat-killed *C. rodentium* stimulations

(Fig S9C). Furthermore, augmented levels of PRMT5 in HEK293 ft cells was sufficient to promote CLCA1 abundance (Fig S10A–C), suggesting that the regulation of CLCA1 by PRMT5 is not restricted to colonic epithelial cells.

In contrast, no significant alterations to the noncanonical goblet cell or icGC programs were found in the *Prmt5*⁺/⁻ colon. WT and *Prmt5*⁺/⁻ colonic epithelium expressed similar levels of *Spdef*, the master transcription factor required for icGC differentiation (Fig S11A). Consistent with these results, WT and *Prmt5*⁺/⁻ mice showed similar weight changes when challenged on the DSS–induced model of colitis, which has been reported to be dependent on icGCs (Fig S11B). Together, these results suggest that PRMT5 preferentially regulates conventional crypt-residing goblet cell gene programs in the colon.

PRMT5 catalyzes the symmetrical dimethylation of arginine residues on target proteins (21). Global symmetric dimethyl arginine levels in *Prmt5*⁺/⁻ colonic epithelial whole cell lysate

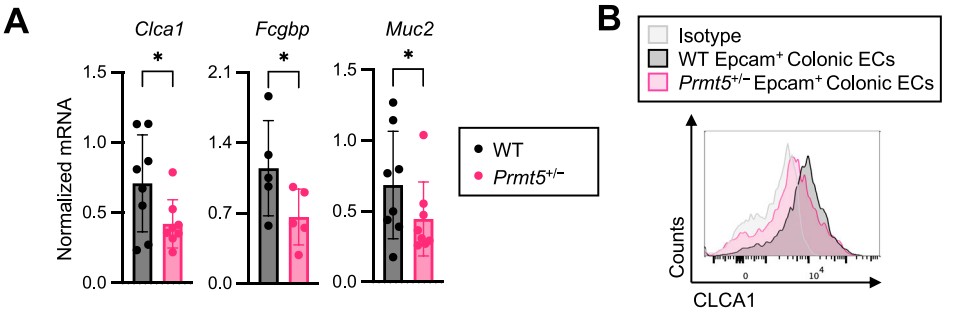

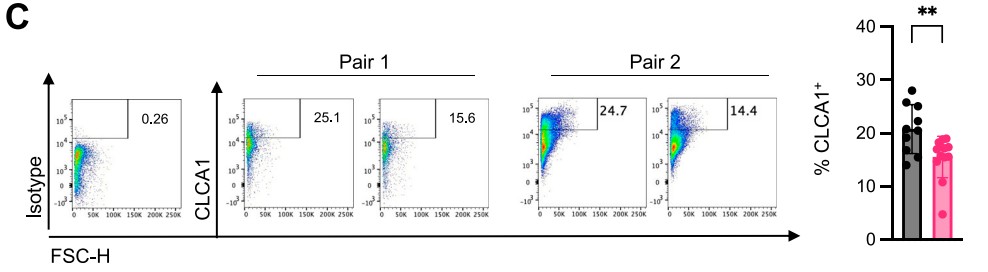

**Figure 5. PRMT5 regulates mucus production and assembly in the colon.**
**(A)** Normalized mRNA expressions of select genes from independent pairs of WT (n = 5 or 8) and $Prmt5^{+/-}$ (n = 5 or 8) colonic IECs. Each dot represents the result from one mouse. Data shown are means ± SD. *$P < 0.05$ ($t$ test). **(B)** Representative histograms of isotype or CLCA1 fluorescent intensity in colonic epithelial cells (ECs) from one pair of WT and $Prmt5^{+/-}$ littermates. **(C)** Left: representative flow cytometry analysis of CLCA1 protein levels in colonic epithelial cells (Epcam⁺) from two pairs of WT and $Prmt5^{+/-}$ littermates. Right: summary of the % of CLCA1⁺ colonic epithelial cells in steady state WT and $Prmt5^{+/-}$ littermates. Each dot represents the result from one mouse. Data shown are means ± SD. **$P < 0.01$ ($t$ test). **(D)** Left: representative images of WGA staining of WT and $Prmt5^{+/-}$ colonic sections. Right: average WGA thickness and fluorescent intensity from WT and $Prmt5^{+/-}$ littermates. Each dot on the graphs represents the average measurement taken from a region of interest within one colonic section. Data shown are means ± SD. **$P < 0.01$, n.s., not significant ($t$ test). Scale bar represents 100 $\mu m$.

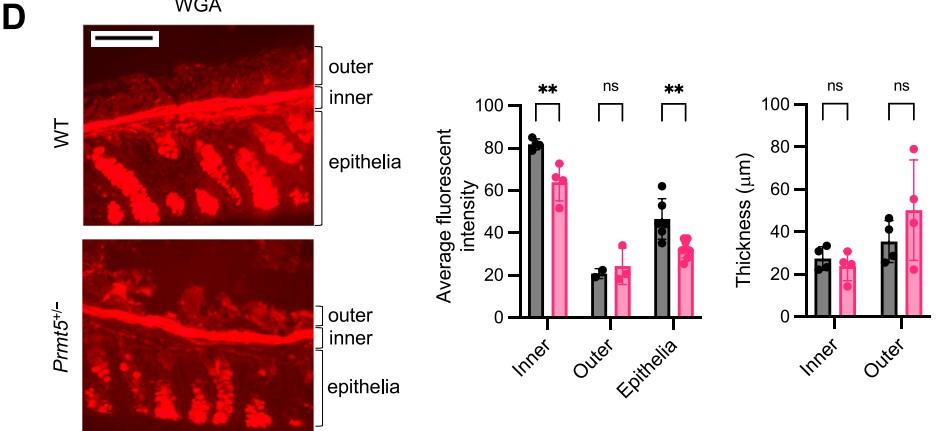

were reduced compared with those found in control cells (Fig S12A). To test whether PRMT5-mediated regulation of goblet cell–related genes is methyltransferase activity-dependent in colonic epithelial cells, we used an organoid culture system free from gut microbiota or immune cells (Fig S12B). Colonic crypts containing epithelial stem cells harvested from WT mice were cultured ex vivo. Organoids were treated concurrently with IL-13 to promote specification towards the secretory lineages, and EPZ015666, a potent and selective inhibitor of PRMT5 enzymatic activity (22), for 12 h (Fig S12B and C). qRT-PCR analysis revealed a dose-dependent reduction of $Clca1$ expression in EPZ015666-treated organoids (Fig S12D). In contrast, $Fcgbp$ and $Muc2$ transcripts were not regulated in a PRMT5 methyltransferase inhibitor–dependent manner. Together, these results suggest that PRMT5 regulation of $Clca1$ is arginine methyltransferase

activity–dependent and cell-intrinsic in the absence of inputs from the gut microbiota or the immune system.

CLCA1 is a metalloprotease implicated in mucus layer organization, and MUC2 and FCGBP are structural building blocks of the intestinal mucus (23). Therefore, we hypothesized that reduced expression of $Clca1$, $Muc2$, and $Fcgbp$ in the $Prmt5^{+/-}$ colon would result in reduced mucus production and alteration in the layered mucus structure. To test this hypothesis, WT and $Prmt5^{+/-}$ colonic sections were stained with WGA, which binds to sialic acid and N-acetylglucosamine residues present on mucins (24). Although the inner and outer mucus layer thicknesses in WT and $Prmt5^{+/-}$ colons were similar, the WGA fluorescent intensities in the $Prmt5^{+/-}$ inner mucus and epithelia were lower than those found in WT mice (Fig 5D). Together, these results suggest that PRMT5 regulates colonic mucus production and inner mucus layer assembly.

## Discussion

Recent studies implicate PRMT5 in colorectal cancer pathogenesis ([25], [26]), but its physiological roles under homeostasis and enteric pathogen challenges are not known. In this study, we found that *Prmt5* is up-regulated in the colon in settings of IBD in humans and *C. rodentium* infection in mice. Interestingly, heat-killed *C. rodentium* and cytokine stimulations from SAA1, IL1β, or IL23 alone were not sufficient to induce *Prmt5* expression in murine-cultured epithelial cells. We suspect that the *C. rodentium*–mediated up-regulation of *Prmt5* may be driven by a more complex mechanism involving non-epithelial cells and is an area of ongoing research. Mice with haploinsufficient levels of *Prmt5* showed a diminished capacity to clear the infection and consequently sustained greater tissue injury. This demonstrates a new role of PRMT5 in inflammatory diseases and host defense against enteric infection.

We uncovered that PRMT5 regulates select canonical crypt goblet cell gene programs, including those involved in anti-microbial responses, such as *Zg16* and *Ang4*, and mucus assembly, such as *Muc2*, *Fcgbp*, and *Clca1*. Interestingly, only a portion of the PRMT5-regulated programs depends on its methyltransferase activity, including *Clca1*, whereas other goblet cell genes, such as *Muc2* and *Fcgbp*, were regulated by PRMT5 in a methyltransferase activity–independent manner. Because of the limited number of primary goblet cells present in the murine colon and organoid culture, future advances in single-cell biochemical and genomic technologies will be needed for investigating the role of PRMT5 in modulating the global chromatin and epigenetic landscape in these cells.

The dense inner mucus layer adjacent to the epithelium is impenetrable to most bacteria, and the loosely organized outer mucus layer is more accessible to the gut microbes and frequently shed into the fecal stream ([27], [28]). In the PRMT5 haploinsufficient colon, deficiency in the canonical crypt goblet cell gene program correlates with aberrant inner mucus assembly. Therefore, we speculate that PRMT5 may protect against enteric pathogen challenge in part by promoting mucus layer assembly, thereby limiting microbial access to the host epithelium tissues and facilitating expedited expulsion of microbes into the fecal stream (modeled in Fig S13). Future microscopy studies will be needed to assess these possibilities. Development of goblet cell subset–specific *Prmt5* knock-out mice and in vivo rescue experiments will also be needed to determine which PRMT5-dependent gene(s) in the canonical crypt goblet cell gene program is essential for host defense against *C. rodentium*. In settings of IBD and barrier disruption, *Prmt5* may be up-regulated as a protective mechanism to promote anti-microbial peptide production and facilitate mucosal barrier repair. In summary, PRMT5 is an important regulator of colonic mucosal defense programs and may have therapeutic potential for combating intestinal inflammatory diseases.

## Materials and Methods

### Mice

*Prmt5*⁺/⁻ mice were obtained from the Wellcome Trust Sanger Institute (MGI ID: 4432368) and crossed to WT C57BL6 mice (JAX) to generate adult WT and *Prmt5*⁺/⁻ littermates for experiments. Adult mice at least 8 wk old were used. All animal studies were approved and followed the Institutional Animal Care and Use Guidelines of the University of California San Diego. Our vivarium at UC San Diego is kept under specific pathogen–free conditions. Regular serology and PCR tests are used to monitor and ensure the absence of epizootic diarrhea of infant mouse virus (EDIM), mouse hepatitis virus (MHV), mouse parvovirus (MPV), minute virus of mice (MVM), Theiler's murine encephalomyelitis virus (TMEV), fur mites, and pinworms.

### Intestinal permeability assay

Mice were deprived of food and bedding for 4 h before oral gavage with 4 kD fluorescein isothiocyanate dextran (FITC-dextran, 100 mg/kg; Millipore-Sigma). Blood samples were taken 4 h post-gavage by submandibular bleed. The FITC-dextran signals were measured using a TECAN fluorescent plate reader at the excitation/emission wavelengths of 485/535 nm.

### *C. rodentium* infection

*C. rodentium* (strain DBS100) cultures were grown overnight in Luria broth (LB), washed, and resuspended in PBS. Male and female mice at least 8 wk old were orally gavaged with $2 \times 10^9$ *C. rodentium* in 0.2 ml PBS. For quantification of *C. rodentium* burden, fecal pellets were collected, homogenized in 1 ml PBS, and clarified at 9,391*g* for 1 min. Supernatants were serially diluted in LB, plated onto Mac-Conkey agar plates, and incubated at 37°C for 20 h. Colony-forming units (CFU) were normalized to fecal pellet weights (per gram). On days 13 and 16, colons were collected for histology.

### DSS-induced colitis

DSS salt colitis grade 36,000–59,000 MW (MP Biomedicals) was added to the drinking water at a final concentration of 2% (wt/vol) and administered for 7 d. Mice were weighed at the start of DSS administration and every day beginning 4 d after the start of DSS administration until day 14.

### Cell culture and transfection

MC38 cells were cultured in DMEM media (Gibco) supplemented with 10% FBS, 2 mM GlutaMAX (Gibco), 1 mM sodium pyruvate (Gibco), and 50 U penicillin–streptomycin. Cells were treated with SAA1 (20 ng/ml; R&D Systems), IL-23 (20 ng/ml; R&D Systems), IL-1β (20 ng/ml; Peprotech), heat-killed *C. rodentium* (Multiplicity of infection = 20), or vehicle (PBS for SAA1, IL-23, and *C. rodentium* treatments; PBS + 0.1% BSA for IL-1β treatment) for 12 h and harvested for total RNA extraction (QIAGEN) and qRT-PCR analysis. For preparation of heat-killed *C. rodentium*, a culture was grown overnight in LB, washed twice and resuspended in PBS, and heat inactivated at 60°C for 30 min.

HEK293 ft cells cultured in 24-well plates containing 250 μl of DMEM supplemented with 10% FBS and 2 mM GlutaMAX (Gibco) were transfected with 0.5 μg of empty or PRMT5-encoding

pcDNA3.1 constructs together with 0.05 μg of pmaxGFP vector (Amaxa) using Lipofectamine 3000 (Invitrogen). Live and GFP[high]-transduced cells were analyzed by flow cytometry 72 h post-transfection.

Colonic crypts were isolated according to the manufacturer's recommendation (technical bulletin #28223; STEMCELL). Briefly, colons were harvested from 6–8 wk-old WT mice and cut into 2 mm pieces. After 20 washes in cold PBS, tissues were resuspended in 25 ml room temperature Gentle Cell Dissociation Reagent (#07174; STEMCELL) and incubated at room temperature for 15 min on a rocking platform at 20 rpm. The pellets enriched with intestinal crypts were resuspended in cold PBS containing 0.1% BSA. Isolated colonic crypts were embedded in Corning Matrigel Matrix (356231; Corning) and seeded onto pre-warmed, non-treated 24-well plates (CytoOne by StarLab) and overlaid with conditioned media (#6005; STEMCELL) as described previously (29). Two days post passage, organoids were treated with recombinant mouse IL-13 (20 ng/ml; BioLegend) and DMSO (Sigma-Aldrich) or EPZ015666 (1–5 μM; Medchemexpress) for 12 h.

### Plasmids

The pcDNA3.1 wildtype murine *Prmt5* overexpression construct was generated using the NEBuilder HiFi DNA Assembly Cloning Kit (New England BioLabs) and primers designed by the NEBuilder Assembly Tool v2.7.1 with NotI and EcoRI extensions. The amplified PCR products were ligated to the NotI- and EcoRI-linearized pcDNA3.1 vector (Invitrogen).

### Flow cytometry analysis

Intestinal epithelial cells were harvested as previously described (30), stained with the LIVE/DEAD Fixable Cell stain (L34957; Thermo Fisher Scientific), fluorescently conjugated antibodies against EpCAM (1:400), and TCRβ (1:400) for 30 min. Cells were fixed/permeabilized (Cat: 00-5521-00; Thermo Fisher Scientific), then incubated with the anti-CLCA1 antibody (ab180851; 1:131 in 1X Permeabilization buffer; Abcam) for 1 h in 4°C, followed by incubation with anti-rabbit IgG (H + L) Cross-Adsorbed Secondary Antibody (1:400; Invitrogen) for 1 h in 4°C. Fixed and permeabilized MC38 cells (Cat: 554714; BD) were stained with the anti-CLCA1 antibody for 1 h in 4°C, followed by anti-rabbit IgG (H + L) Cross-Adsorbed Secondary Antibody for 1 h in 4°C. Lastly, cells were incubated with PE-conjugated anti-PRMT5 antibody (ab210437; 1:100 in 1X Permeabilization; Abcam) for 1 h in 4°C. Fixed and permeabilized HEK293 ft cells (Cat: 554714; BD) were incubated with anti-CLCA1 antibody, anti-GFP antibody (A-21311; 1:800 in 1X Permeabilization buffer; Invitrogen), anti-rabbit IgG (H + L) Cross-Adsorbed Secondary Antibody, and anti-PRMT5 antibody for 1 h in 4°C. Lamina propria lymphocytes were isolated from a 40:80% Percoll gradient interphase. Cells were stimulated with 5 ng/ml PMA (Millipore Sigma) and 500 ng/ml ionomycin (Millipore Sigma) or IL-23 (20 ng/ml) in the presence of GolgiStop (BD Bioscience) for 4 h at 37°C, followed by cell surface marker staining. Fixation/permeabilization buffers (eBioscience and BD) were used per manufacturer instructions to assess intracellular transcription factor and cytokine expression. Antibodies are listed in Table S2. Flow cytometry data were analyzed with FlowJo (version 10.9.0).

### Western blot analysis

For whole cell lysates, cells were lysed in 25 mM Tris, pH 8.0, 100 mM NaCl, and 0.5% NP40 with protease inhibitors for 30 min on ice. Samples were spun down at 14,000*g* for 10 min, and soluble protein lysates were harvested. 20–50 μg proteins were loaded on each lane. Blots were blocked with the Odyssey Blocking Buffer (Li-CoR Biosciences) and probed with primary antibodies listed in Table S2. Following incubation with respective IRDye secondary antibodies (Li-CoR Biosciences), infrared signals on each blot were captured by the Li-CoR Odyssey CLX.

### cDNA synthesis and qRT-PCR

Total RNA was extracted with the RNeasy Plus kit (QIAGEN) and reverse transcribed using the SuperScript III First-Strand Synthesis System (Invitrogen). Real-time RT–PCR (qRT-PCR) was performed using iTaq Universal SYBR Green Supermix (Bio-Rad). For primary IECs, results were normalized to mouse *Gapdh*. For MC38 cells, results were normalized to mouse *Hprt*. Primers were designed using Primer-BLAST to span across splice junctions, resulting in PCR amplicons that span at least one intron. Primer sequences are listed in Table S3.

### WGA, IHC, PAS, and H&E staining

To preserve the mucus layer for WGA staining, colonic samples were unflushed and fixed in Carnoy's fixative consisting of 60% ethanol, 30% chloroform, and 10% glacial acetic acid for 1 h and preserved in 80% ethanol overnight. Fixed tissues were embedded in paraffin. 5 μm deparaffined sections were stained with Alexa Fluor 594 conjugated WGA (1:500; Invitrogen) overnight to detect N-acetyl-D-glucosamine and N-acetyl-D-neuraminic acid residues on mucins. VECTASHIELDR (Vector Laboratories) mounting medium containing DAPI was used to visualize nuclei. Stained sections were imaged by fluorescent microscopy. All samples were analyzed using NIH ImageJ. Randomly chosen regions of interest in each colonic section were quantified for inner and outer mucus layer thicknesses and fluorescent intensities. The average quantification of measurements from each randomly chosen region was included in the final graph.

For PAS or immunohistochemical staining, colonic tissues were fixed overnight in 10% formalin (Research Products International) at room temperature. Paraffin-embedded tissues were sectioned into 5 μm slices. Staining was then performed with either negative control IgG antibody, anti-DCLK1 (1:4,000) antibody (Abcam), or anti-PRMT5 (1:600) antibody (Millipore Sigma) overnight in a humid chamber at 4°C. The next day, sections were washed with TBST and then sequentially overlaid with biotinylated goat anti-rabbit (111-065-045; Jackson ImmunoResearch) at 1:500, followed by HRP-labelled streptavidin (16-030-084; Jackson ImmunoResearch) at 1:500. Substrate was then overlaid with 3-amino-9-ethylcarbazole from Vector laboratories following manufacturer's directions for 30 min followed by nuclear counterstain with Mayer's haematoxylin. Images were acquired using the AT2 Aperio Scan Scope (UCSD Moores Cancer Center Histology Core). Three intestinal regions per tissue image were randomly selected for QuPath analysis. DCLK1[+] tuft cells were determined by QuPath positive cell detection (31)

(minimum area = 10 $\mu$m$^2$, maximum area = 400 $\mu$m$^2$, intensity threshold = 0.4). Mucin$^+$ goblet cells were assessed similarly (minimum area = 40 $\mu$m$^2$, maximum area = 800 $\mu$m$^2$, intensity threshold = 0.6). The average score from three regions examined in each tissue was included in the final graph.

For histopathological analysis, distal colons were opened longitudinally, rinsed with PBS, and fixed overnight at room temperature in 10% formalin (Fisher Chemicals). Three random regions on each colonic tissue were scored in a double-blind fashion as described previously (32, 33), and the averages of the three scores were graphed for each tissue.

### RNA-seq

Colonic epithelial cell RNAs from two pairs of gender-matched and cohoused steady-state WT and *Prmt5*$^{+/-}$ mice were used to generate sequencing libraries. 100 bp paired-end sequencing was performed on an Illumina HiSeq4000 by the Institute of Genomic Medicine at the University of California San Diego. Each sample yielded ~30–40 million reads. Paired-end reads were aligned to the mouse mm10 genome with the STAR aligner version 2.6.1a (34) using the parameters: "--outFilterMultimapNmax 20 --alignSJoverhangMin 8 --alignSJDBoverhangMin 1 --outFilterMismatchNmax 999 --out-FilterMismatchNoverReadLmax 0.04 --alignIntronMin 20 --alignIntronMax 1000000 --alignMatesGapMax 1000000." Uniquely mapped reads overlapping with exons were counted using featureCounts (35) for each gene in the GENCODE.vM19 annotation. Differential expression analysis was performed using DESeq2 (v1.18.1 package) (36), including a covariate in the design matrix to account for differences in harvest batch/time points. Regularized logarithm (rlog) transformation of the read counts of each gene was carried out using DESeq2. Differentially expressed protein-coding genes with minimal counts of 10, log$_2$ fold change cutoffs of greater or equal to 0.5 or lesser or equal to –0.5, and *P*-values < 0.05 were considered significant. Gene ontology analyses were performed on differentially expressed genes with a cut-off of log$_2$ fold change greater or equal to 1 or lesser or equal to –1. Gene set enrichment analysis (GSEA) was carried out using the pre-ranked mode of the GSEA software with default settings (37). The gene list from DEseq2 was ranked by calculating a rank score of each gene as –log$_{10}$(*P*-value) × SIGN(log$_2$[Fold Change]), in which SIGN gives the sign of the number and Fold Change is the fold change of gene expression in *Prmt5*$^{+/-}$ over that found in WT.

Human colonic epithelium datasets analyzed were obtained from GSE179128 and analyzed using StepMiner. The expression levels of all genes in these datasets were converted to binary values (high or low) using the StepMiner algorithm (38, 39) which undergoes an adaptive regression scheme to verify the best possible up and down steps based on sum-of-square errors. The steps are placed between time points at the sharpest change between expression levels which gives us the information about timing of the gene expression–switching event. To fit a step function, the algorithm evaluates all possible steps for each position and computes the average of the values on both sides of a step for the constant segments. An adaptive regression scheme is used that chooses the step positions that minimize the square error with the fitted data. Finally, a regression test statistic is computed as follows:

$$F\,stat = \frac{\sum_{i=1}^{n}\left(\hat{X}_i - \overline{X}\right)^2 \Big/ (m-1)}{\sum_{i=1}^{n}\left(X_i - \hat{X}_i\right)^2 \Big/ (n-m)}$$

where $X_i$ for $i$ = 1 to $n$ are the values, $\hat{X}_i$ for $i$ = 1 to $n$ are fitted values. m is the degrees of freedom used for the adaptive regression analysis. $\overline{X}$ is the average of all the values: $\overline{X} = \frac{1}{n} * \sum_{j=1}^{n} X_j$. For a step position at k, the fitted values $\hat{X}_l$ are computed by using $\frac{1}{k} * \sum_{j=1}^{n} X_j$ for $i$ = 1 to $k$ and $\frac{1}{(n-k)} * \sum_{j=k+1}^{n} X_j$ for $i$ = $k$+1 to $n$. Gene expression values were normalized according to a modified Z-score approach centered around *StepMiner* threshold (formula = (expr - SThr)/3*stddev). The normalized expression values for all genes were added together to create the final score for the gene signature. The samples were ordered based on the final signature score. Classification of sample categories using this ordering is measured by ROC–AUC (receiver operating characteristics area under the curve) values. Welch's two sample *t* test (unpaired, unequal variance [equal_var = False], and unequal sample size) parameters were used to compare the differential signature score in different sample categories. Violin plots are created using Python Seaborn package version 0.10.1. Expression patterns of the genes that are highly expressed in different groups are unbiasedly clustered based on their modified Z-score approach centered around *StepMiner* threshold, in all the samples using the Seaborn clustermap package (v 0.12) in Python.

### Statistical analysis

All values are presented as mean ± SD. Significant differences were evaluated using GraphPad Prism 9 software (GraphPad). The *t* test was used to determine significant differences between two groups with normal distribution. A two-tailed *P*-value of < 0.05 was considered statistically significant in all experiments.

## Data Availability

All data needed to evaluate the conclusions in the study are present in the article and/or the Supplementary Materials. Raw sequencing data for the RNA-seq experiment are available at Gene Expression Omnibus under accession number GSE221566.

## Supplementary Information

## Acknowledgements

JE Hernandez, S Ma, S Steele, Z Xiao, C-J Lai, and WJM Huang are partially funded by the Edward Mallinckrodt, Jr. Foundation, and the National Institutes of Health R01-GM124494 (to WJM Huang) and T32 CA067754 (to JE Hernandez). B Schnabl is supported by NIH grants R01 AA24726, R37

AA020703, U01 AA026939, U01 AA026939-04S1, by Award Number BX004594 from the Biomedical Laboratory Research and Development Service of the VA Office of Research and Development, and a Harrington Discovery Institute Foundation Grant (to B Schnabl) and services provided by NIH centers P30 DK120515 and P50 AA011999. C Llorente is supported by NIH grants R01 AA029106-01A1, D34HP31027, Pilot and Feasibility grants from P30 DK120515 and 5P50AA011999, and by the 8998GA Pinnacle Research Award from the AASLD. S Sinha and P Ghosh are partially supported by UG3TR003355, R01 AI141630, and R01 AI155696 (to P Ghosh) and by the Leona M. and Harry B. Helmsley Charitable Foundation. S Sinha is supported by the American Association of Immunologists Intersect Fellowship Program for Computational Scientists and Immunologists.

## Author Contributions

JE Hernandez: conceptualization, data curation, formal analysis, and writing—original draft, review, and editing.
C Llorente: resources, formal analysis, methodology, and writing—review and editing.
S Ma: resources, methodology, and writing—review and editing.
KT Miyamoto and P Ghosh: resources, data curation, formal analysis, and writing—review and editing.
S Sinha and S Steele: data curation, formal analysis, and writing—review and editing.
Z Xiao and C-J Lai: formal analysis, methodology, and writing—review and editing.
EI Zuniga: conceptualization, resources, supervision, and writing—review and editing.
B Schnabl: resources, supervision, and writing—review and editing.
WJM Huang: conceptualization, supervision, project administration, and writing—review and editing.

## Conflict of Interest Statement

B Schnabl has been consulting for Ambys Medicines, Ferring Research Institute, Gelesis, HOST Therabiomics, Intercept Pharmaceuticals, Mabwell Therapeutics, Patara Pharmaceuticals, and Takeda. B Schnabl is founder of Nterica Bio. UC San Diego has filed several patents with B Schnabl as inventor. B Schnabl's institution UC San Diego has received research support from Artizan Biosciences, Axial Biotherapeutics, BiomX, CymaBay Therapeutics, NGM Biopharmaceuticals, Prodigy Biotech, and Synlogic Operating Company. All other authors declare no competing financial interests.

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
