## [Reviewer comments · Life Science Alliance]

Life Science Alliance

The arginine methyltransferase PRMT5 promotes mucosal defense in the intestine

Wendy Huang, Juan Hernandez, Cristina Llorente, Shengyun Ma, Kiana Miyamoto, Saptarshi Sinha, Scarlet Steele, Zihui Xiao, Ching-Jung Lai, Elina Zuniga, Pradipta Ghosh, and Bernd Schnabl

DOI: <https://doi.org/10.26508/lsa.202302026>

Corresponding author(s): Wendy Huang, University of California, San Diego

Review Timeline:	Submission Date:	2023-03-07
	Editorial Decision:	2023-04-14
	Revision Received:	2023-08-01
	Editorial Decision:	2023-08-22
	Revision Received:	2023-08-27
	Accepted:	2023-08-28

Scientific Editor: Novella Guidi

Transaction Report:

April 14, 2023

Re: Life Science Alliance manuscript #LSA-2023-02026

Dr. Wendy Jia Men Huang
University of California, San Diego
9500 Gilman Drive MC0651
George Palade Laboratories, Room 321B
La Jolla 92093

Dear Dr. Huang,

Thank you for submitting your manuscript entitled "The arginine methyltransferase PRMT5 promotes mucosal defense in the intestine" to Life Science Alliance. The manuscript was assessed by expert reviewers, whose comments are appended to this letter. We invite you to submit a revised manuscript addressing the Reviewer comments.

Thank you for this interesting contribution to Life Science Alliance. We are looking forward to receiving your revised manuscript.

Sincerely,

B. MANUSCRIPT ORGANIZATION AND FORMATTING:

Reviewer #1 (Comments to the Authors (Required)):

In this report, Hernandez et al. reveal that PRMT5 in the murine colon is upregulated in response to *Citrobacter rodentium* infection. They show that pathogen clearance in mice with haploinsufficient expression of PRMT5 is significantly delayed compared to wildtype littermate controls. They also found potential downstream target genes of PRMT5 through expression profiling analysis in this context and proposed that PRMT5 is a novel regulator of mucosal defense. Overall, experiments were well designed and the manuscript was nice structured. However, based on current claims from the authors, it is not sufficient to draw the conclusions from experiments authors conducted. Major points are listed below:

1. The logic about the role of PRMT5 during *Citrobacter rodentium* infection is difficult to understand. Given that PRMT5 in the murine colon is upregulated in response to *Citrobacter rodentium* infection, would upregulated PRMT5 help clear the infection later? Can authors explain why and how *Citrobacter rodentium* infection upregulates PRMT5 specifically?
2. The authors claims that PRMT5 regulates some key genes, such as *Clca1*, in the context of infection. The use of EPZ01566 in organoid culture further suggests that the regulation is methyltransferase activity dependent. However, in order to confirm this result, the authors need to perform experiments with methyltransferase activity-dead PRMT5 mutant vs wildtype at least in organoid cultures.
3. If *Clca1*, *Muc2*, and *Fcgbp* are key downstream genes of PRMT5 in response to *Citrobacter rodentium* infection, can authors show rescued phenotype of *Prmt5*^{+/-}-mice with enforced expression of *Clca1*, *Muc2*, or *Fcgbp* by performing *Citrobacter rodentium* clearance experiments?
4. *Clca1*, *Muc2*, and *Fcgbp* are key downstream genes of PRMT5. It would be more convincing to include their protein levels by western blot analysis in the manuscript.
5. The potential mechanism by which PRMT5 promotes the expression of *Clca1*, *Muc2*, and *Fcgbp* is very interesting. Can authors expect which histone mark, either H4R3me2s or H3R8me2s is mainly involved in the process? Or totally different Epi-null process? Some experimental clues would be very helpful.

Reviewer #2 (Comments to the Authors (Required)):

Summary

This is a very interesting and well written manuscript. The research study uses the murine model of *Citrobacter rodentium* infection-induced colitis to investigate the role of PRMT5 in inflammatory bowel disease (IBD) affecting humans. Herein, the paper uncovers that mice with lower levels of PRMT5 were susceptible to infection and suffered increased tissue damage. Using RNA sequencing, the paper also found that PRMT5 controls some anti-microbial and mucous assembly genes programs in the colon, but does not affect the master transcription factor controlling the growth of the colon's crypts. Overall, this paper provides novel insights into the role of PRMT5 in protecting against intestinal infections and may serve as a therapeutic target for human IBD.

Minor suggestion

-Page 7, Line 5 has *pmrt5* instead of PRMT5

Reviewer #3 (Comments to the Authors (Required)):

Hernandez JE et al present experimental data supporting a promoting role of the type II arginine methyl transferase gene in mucosal defense in the intestine. Experiments rely on the use of *Prmt5*^{+/-} mice as the full deficiency is lethal.

The main arguments supporting their claim are

- The overexpression of *Prmt5* in colons from active IBD patients or *C rodentium*-infected mice
- A modest increase in the delay for *C rodentium* clearance but no difference in the susceptibility to DSS colitis
- A transcriptional IEC profile showing a reduced expression of gene sets linked to goblet and tuft cell gene programs
- Preliminary results showing at the protein level that some effector proteins such as *Muc2* or *Clca1* show a reduced expression in steady-state *Prmt5*^{+/-} colons

Generally, the experimental approach is reasonable but the observed phenotypes are modest and sometimes at the limit of statistical significance.

More specifically:

1: Figure 1 shows evidence for delayed clearance of *C. rodentium* in the feces. Apparently, the results shown derive from a single experiment with 12-13 mice in each experimental condition. Both fecal CFU/g and histological scoring are at the limit of significance. The histology scoring was only performed on 3 to 4 mice. Since this is a major phenotype of this paper, it would have been important to strengthen these results and show their reproducibility in independent experiments.

Furthermore, since the clearance of *C. rodentium* depends both on the quality of the mucosal barrier and on the presence of specific immune cells such as ILCs and their cytokines, it would be useful to score these parameters.

2: Figure 2 shows IEC RNAseq data obtained from steady-state colons and identify putative *Prmt5*-regulated genes. Have the authors tested their expression by qRT-PCR in infected mice? since *Prmt5* expression is boosted by infection, the differential expression of target genes could be much more significant.

3: Figure 3 shows convincing GSEA data confirming the contribution of *Prmt5* to the developmental programs of goblet and tuft cells. As raised in the paper, this could be due to a reduced proportion of these cell subsets or to the downregulation of specific genesets among lineage-committed cells. The authors favor the second hypothesis and show histology scores quantifying PAS+ and DLCK1 (Fig S3) cells on sections of colonic mucosa. Here again, the results are obtained from the analysis of a small number of mice (3 to 5). Furthermore, concerning PAS+ staining, the analysis of the Figure 4A tends to suggest that whereas the proportion of PAS+ cells is comparable at the top of the crypts (enlarged image), the situation might be different if one considers the whole crypt length. It would be important to demonstrate this point by using complementary methods.

4: The last part of the manuscript investigates the expression of specific proteins linked to the function of goblet cells such as CLCA1 and mucus-associated proteins. Results show modest differences with significant inter-individual variability. More specifically in Figure 5C the quantification of Epcam+ CLCA1+ cells raise a concern since in the control cohort the statistical difference is only driven by 3 mice whereas other control mice show the same result as that of *Prmt5* +/- mice.

This type of comment also applies to the quantification of mucus using WGA staining. The analysis is taken from "a region of interest within one colonic section". The quantification is performed using the average fluorescent intensity on these colonic sections. These results would also be strengthened by using different types of staining methods or western blot analysis. Since the authors have access to organoid cultures and used it to quantify *Clca1* expression, why not use this methodology on *Prmt5* +/- mice and score different parameters to quantify goblet (or tuft) cell maturation and the expression of various effector proteins.

5: Finally, from a mechanistic point of view, *Prmt5* will contribute to the asymmetric methylation of arginine on proteins such as histones. Have the authors attempted to quantify the accessibility of the promoter of candidate target genes such as *Clca1* or *Muc2*? This type of experiment would help demonstrating their point.

In conclusion, this manuscript unravels a putative role of *Prmt5* in the control of expression of functional gene programs within specific IEC. While the effects of *Prmt5* haploinsufficiency are modest in their experimental setting, these results are of interest but would need to be further supported. Organoids from *Prmt5* +/- mice could be better exploited for their demonstration.

Detailed responses to Reviewers' comments

Reviewer #1:

In this report, Hernandez et al. reveal that PRMT5 in the murine colon is upregulated in response to *Citrobacter rodentium* infection. They show that pathogen clearance in mice with haploinsufficient expression of PRMT5 is significantly delayed compared to wildtype littermate controls. They also found potential downstream target genes of PRMT5 through expression profiling analysis in this context and proposed that PRMT5 is a novel regulator of mucosal defense. Overall, experiments were well designed and the manuscript was nice structured. However, based on current claims from the authors, it is not sufficient to draw the conclusions from experiments authors conducted. Major points are listed below:

Our response: We thank Reviewer 1 for finding our experiments well-designed and manuscript well-structured. Reviewer 1's input has helped us improve our manuscript.

1. The logic about the role of PRMT5 during *Citrobacter rodentium* infection is difficult to understand. Given that PRMT5 in the murine colon is upregulated in response to *Citrobacter rodentium* infection, would upregulated PRMT5 help clear the infection later?

Our response: In the revised Supplementary Figure S13A, we included a model figure to better illustrate our findings that the upregulation of PRMT5 ensures a robust production of mucus assembly factors, such as CLCA1 and FCGBP, which likely contributes to reducing microbial access to the host epithelium tissues and expediting microbe elimination into the fecal stream.

Can authors explain why and how *Citrobacter rodentium* infection upregulates PRMT5 specifically?

Our response: We thank Reviewer 1 for this suggestion. Our initial hypothesis was that *Citrobacter rodentium* infection upregulates PRMT5 by either directly stimulating intestinal epithelial cells or indirectly by the immune modulatory cytokines released by tissue-resident immune cells surveying the colonic tissue. Therefore, we tested whether treatment of a murine colonic epithelial cell line (MC38) with heat-killed *C. rodentium* or recombinant immune modulatory cytokines known to be induced in mouse models of bacterial enteric infection, including SAA1, IL-23, and IL1 β (Omenetti et al., 2019, *Immunity* 51, 77–89; Jacobse et al., 2023, *Cell Reports* 42, 112128; Alipour, Misagh, et al. *PloS one* 8.12 (2013): e80656), modulate *Prmt5* gene expression. Interestingly, none of the treatment schemes we tested altered *Prmt5* levels (revised Supplementary Figure S9C). In the revised Discussion section on page 10, we clarified that it remains unclear the exact mechanism underlying how *C. rodentium* upregulates *Prmt5* and this should be the subject of future studies and beyond the scope of the current manuscript.

2. The authors claims that PRMT5 regulates some key genes, such as *Clca1*, in the context of infection. The use of EPZ01566 in organoid culture further suggests that the regulation is methyltransferase activity dependent. However, in order to confirm this result, the authors need to perform experiments with methyltransferase activity-dead PRMT5 mutant vs wildtype at least in organoid cultures.

Our response: As Reviewer 1 suggested, we generated expression constructs encoding wildtype PRMT5 (PRMT5^{WT}), as well as two mutants previously reported to be enzymatically dead in vitro biochemical assays (PRMT5^{E425Q} and PRMT5^{G348A/R349A}) (Hamard *et al.* 2018, Cell Reports 24, 2643-2657; Pal *et al.* Molecular and Cell Biology 23, 7475-7487). We confirmed that transient transfection of the PRMT5^{WT} constructs in HEK293ft cells was sufficient to promote CLCA1 expression (Suppl. Fig. S10A-C) and global arginine symmetric dimethylation (Panel A in Figure below). To our surprise, both the PRMT5^{E425Q} and PRMT5^{G348A/R349A} mutants showed similar ability to potentiate global arginine symmetric dimethylation when transiently expressed in HEK293ft cells (Panel A in Figure below). Unlike the PRMT5^{E425Q} and PRMT5^{G348A/R349A} mutants, treatment of colonic epithelial cells with the PRMT5-selective methyltransferase inhibitor, EPZ015666, was able to significantly reduce global arginine symmetric dimethylation (Panel B in Figure below). Given these results, we employed the EPZ015666 as the primary tool to address whether PRMT5 enzymatic activities were involved in regulating goblet cell gene programs. In the revised Suppl. Fig. S12B, we included RT-qPCR results for *Fcgbp* and *Muc2*, in addition to *Clca1*, to provide examples of goblet cell-related genes that depend on PRMT5 in a methylation-independent and dependent manner.

A. Representative western blots of PRMT5, global symmetric dimethyl-arginine (SDMe-Arg), and βTubulin in whole cell lysates of HEK293 cells transfected with the indicated vectors.
 B. Representative western blots of global SDMe and βTubulin in colon IEC whole cell lysates from WT mice treated with vehicle or EPZ015666 once every 2 days for 6 days.

3. If *Clca1*, *Muc2*, and *Fcgbp* are key downstream genes of PRMT5 in response to *Citrobacter rodentium* infection, can authors show rescued phenotype of *Prmt5*^{+/-} mice with enforced expression of *Clca1*, *Muc2*, or *Fcgbp* by performing *Citrobacter rodentium* clearance experiments?

Our response: We agree with Reviewer 1 that it would be important to perform the proposed *in vivo* rescue experiments and determine which of the PRMT5-dependent gene(s) contribute to the *C. rodentium* phenotype observed in the *Prmt5*^{+/-} mice. However, the generation of transgenic mouse lines to enforce the expression of *Clca1*, *Muc2*, and/or *Fcgbp* in colonic goblet cells will require 6-9 months. Due to our limited time and resources, we are unable to perform these experiments. In the revised Discussion (page 11), we acknowledged the need for future rescue experiments to validate our findings.

4. *Clca1*, *Muc2*, and *Fcgbp* are key downstream genes of PRMT5. It would be more convincing to include their protein levels by western blot analysis in the manuscript.

Our response: We thank Reviewer 1 for this suggestion. In fact, we do have the CLCA1, MUC2, and FCGBP western blot analysis results, see the revised Supplementary Figure S6A. These experiments were performed on total colonic epithelial cell whole cell lysates. These assays revealed that FCGBP protein levels in the PRMT5-deficient tissues were significantly lower than those found in wildtype tissues. In the same assay, MUC2 and CLCA1 levels were modestly reduced in the PRMT5-deficient tissues.

We would like to point out that one important caveat of the Western approach is the inability to exclude any epithelial cells that were dead or dying during the colonic tissue harvest process, which typically takes about 1-2hrs. Therefore, we decided to employ flow cytometry analysis to address this issue, where we can perform live/dead cell staining to restrict our analysis to live epithelial cells (see our gating strategy detailed in Supplementary Figure S7A). At the time of our study, commercially available flow cytometry grade antibodies were only available for detecting CLCA1, and not FCGBP or MUC2. With the ability to exclude dead cells, we confirmed that the proportions of CLCA1-positive live colonic epithelial cells in the PRMT5-deficient tissues were significantly lower than those found in wild-type tissues (original Figure 5C). In the revised Figure 5C, we now included flow cytometry analysis results from additional pairs of WT and PRMT5 heterozygote mice, confirming that PRMT5-deficient colonic epithelial cells showed reduced intracellular CLCA1 protein staining compared to those found in wild-type tissues.

As the commercially available FCGBP and MUC2 antibodies were not suitable for flow cytometry, we, therefore, employed the WGA staining assay as a secondary method to confirm altered mucin levels in the PRMT5-deficient colon (Figure 5D).

5. The potential mechanism by which PRMT5 promotes the expression of *Clca1*, *Muc2*, and *Fcgbp* is very interesting. Can authors expect which histone mark, either H4R3me2s or H3R8me2s is mainly involved in the process? Or totally different Epi-null process? Some experimental clues would be very helpful.

Our response: We thank Reviewer 1 for this suggestion. In an attempt to address these questions, we tested whether it is feasible to employ the Chromatin Immunoprecipitation assay to assess the role of PRMT5 in epigenetic marks deposition on the *Clca1*, *Muc2*, and *Fcgbp* genomic loci. From our ChIP-qPCR experiment using the robust H3K4me3 antibodies (marking active chromatin) on 10 million formaldehyde crosslinked colonic epithelial cells, we were able to observe a robust signal on the housekeeping *Gapdh* gene promoter as expected. However, this assay was unable to pick up signals from the *Clca1* or *Fcgbp* promoters (see figure below). We suspect that this technical challenge is likely due to the limited number of *Clca1* and *Fcgbp*-producing goblet cells present in our sample (less than 10% of the total colonic epithelial cells). We estimated that it will require 30-40 experimental mice per genotype to obtain the sufficient number of purified goblet cells by FACS sorting for this type of assay. In the revised Discussion on page 10, we clarified the current technical challenges for determining whether PRMT5-dependent chromatin modification is involved in its mucosal defense role in the colon.

H3K4me3 enrichment on the *Gapdh*, *Clca1*, and *Fcgbp* promoter from the ChIP-qPCR assay using 10 million colon IECs isolated from WT mice. Each dot represents the result from one mouse. * $P < 0.05$, n.s. not significant (t-test).

Reviewer #2:

Summary

This is a very interesting and well written manuscript. The research study uses the murine model of *Citrobacter rodentium* infection-induced colitis to investigate the role of PRMT5 in inflammatory bowel disease (IBD) affecting humans. Herein, the paper uncovers that mice with lower levels of PRMT5 were susceptible to infection and suffered increased tissue damage. Using RNA sequencing, the paper also found that PRMT5 controls some anti-microbial and mucous assembly genes programs in the colon, but does not affect the master transcription factor controlling the growth of the colon's crypts. Overall, this paper provides novel insights into the role of PRMT5 in protecting against intestinal infections and may serve as a therapeutic target for human IBD.

Our response: We thank Reviewer 2 for finding our work novel and insightful.

Minor suggestion

-Page 7, Line 5 has pmrt5 instead of PRMT5

Our response: We thank Reviewer 2 for this suggestion. On page 7 of the revised text, we have corrected the text to “PRMT5” as suggested.

Reviewer #3:

Hernandez JE et al present experimental data supporting a promoting role of the type II arginine methyl transferase gene in mucosal defense in the intestine. Experiments rely on the use of Prmt5^{+/-} mice as the full deficiency is lethal. The main arguments supporting their claim are

- The overexpression of Prmt5 in colons from active IBD patients or *C. rodentium*-infected mice
- A modest increase in the delay for *C. rodentium* clearance but no difference in the susceptibility to DSS colitis
- A transcriptional IEC profile showing a reduced expression of gene sets linked to goblet and tuft cell gene programs
- Preliminary results showing at the protein level that some effector proteins such as Muc2 or Clca1 show a reduced expression in steady-state Prmt5 ^{+/-} colons

Generally, the experimental approach is reasonable but the observed phenotypes are modest and sometimes at the limit of statistical significance.

Our response: We thank Reviewer 3 for their valuable input which has helped us improve our manuscript.

More specifically:

1: Figure 1 shows evidence for delayed clearance of *C. rodentium* in the feces. Apparently, the results shown derive from a single experiment with 12-13 mice in each experimental condition. Both fecal CFU/g and histological scoring are at the limit of significance.

Our response: In the revised Figure 1E figure legend, we have clarified that the “Graph represents results from 3 independent *C. rodentium* infection experiments combined. The first experiment included WT (n=4) and Prmt5^{+/-} (n=5) mice. The second experiment included WT (n=3) and Prmt5^{+/-} (n=3) mice. The third experiment included WT (n=6) and Prmt5^{+/-} (n=5) mice.”

The histology scoring was only performed on 3 to 4 mice. Since this is a major phenotype of this paper, it would have been important to strengthen these results and show their reproducibility in independent experiments.

Our response: We thank Reviewer 3 for pointing this out. In the revised Figure 1F, we have now included the scores from additional experimental pairs from an independent

experiment, confirming higher pathologies in the *C. rodentium*-infected PRMT5-deficient colon.

Furthermore, since the clearance of *C. rodentium* depends both on the quality of the mucosal barrier and on the presence of specific immune cells such as ILCs and their cytokines, it would be useful to score these parameters.

Our response: We thank Reviewer 3 for these suggestions. In the original Figure 5D, we employed the WGA staining assay to assess mucosal barrier qualities in the WT and *Prmt5*^{+/-} colons. This initial assay revealed a reduction in WGA fluorescent intensity in the *Prmt5*^{+/-} inner mucus layer and epithelia, thereby suggesting alteration of the inner mucus assembly and compromised mucus barrier integrity. In the revised Supplementary Figure S3B, we further assessed the mucosal barrier function of control and PRMT5-deficient mice using the fluorescein isothiocyanate (FITC)-dextran permeability assay. Interestingly, results from the FITC-dextran assay showed that PRMT5 is not involved in regulating the permeability of 4kDa molecules in the intestine. In the revised Supplementary Figure S4A, we examined the cytokine production capacities of the different ILC populations in the colonic lamina propria from *C. rodentium*-infected wildtype and PRMT5-deficient mice and found comparable production potentials of IFN γ , IL-22, and IL-17A, which play crucial roles in protection against *C. rodentium* enteric infection.

2: Figure 2 shows IEC RNAseq data obtained from steady-state colons and identify putative *Prmt5*-regulated genes. Have the authors tested their expression by qRT-PCR in infected mice? Since *Prmt5* expression is boosted by infection, the differential expression of target genes could be much more significant.

Our responses: As suggested, we examined the expression of PRMT5-regulated genes identified from steady-state tissues in *C. rodentium*-infected mice using qRT-PCR. In the revised Supplementary Figure S8A, we showed that *Ctca1* and *Dclk1* transcript abundances were significantly reduced in *C. rodentium*-infected PRMT5-deficient tissues. *Fcgbp*, *Muc2*, and *Ang4* transcript abundances were only slightly, but not significantly, reduced. We would like to point out that interpretation of these results needs to be taken with caution as it may be difficult to discern the effect of PRMT5 from those driven by the elevated presence of *C. rodentium* and higher tissue damage and inflammation in the *Prmt5*^{+/-} colon at the time of harvest.

3: Figure 3 shows convincing GSEA data confirming the contribution of *Prmt5* to the developmental programs of goblet and tuft cells. As raised in the paper, this could be due to a reduced proportion of these cell subsets or to the downregulation of specific genesets among lineage-committed cells. The authors favor the second hypothesis and show histology scores quantifying PAS+ and DLCK1 (Fig S3) cells on sections of colonic mucosa. Here again, the results are obtained from the analysis of a small number of mice (3 to 5). Furthermore, concerning PAS+ staining, the analysis of the Figure 4A tends to suggest that whereas the proportion of PAS+ cells is comparable at the top of the crypts (enlarged image), the situation might be different if one considers

the whole crypt length. It would be important to demonstrate this point by using complementary methods.

Our responses: We thank Reviewer 3 for pointing this out. We would like to clarify that our initial intent was to use the image enlargement window in the original Figure 4A to provide readers with a close-up view of the PAS-stained tissue, and this was not meant to represent the regions we quantified. In the revised Figure 4A, we now included boxes with dotted lines to depict the exact regions we scored (whole crypt length). We have also performed additional PAS scoring on additional WT (total n=7) and *Prmt5*^{+/-} (total n=9) mice, confirming comparable mucin+ goblet cell numbers, crypt numbers, and crypt height in the wildtype and *Prmt5*^{+/-} cohoused gender matched littermates.

In revised Supplementary Figure S5A, we have also included boxes with dotted lines to depict the exact regions we scored (whole crypt length) for the DCLK1 IHC analyses. We now included scores from additional WT (total n=5) and *Prmt5*^{+/-} (total n=5) mice, confirming a comparable DCLK1+ tuft cell number in these mice.

4: The last part of the manuscript investigates the expression of specific proteins linked to the function of goblet cells such as CLCA1 and mucus-associated proteins. Results show modest differences with significant inter-individual variability. More specifically in Figure 5C the quantification of Epcam+ CLCA1+ cells raise a concern since in the control cohort the statistical difference is only driven by 3 mice whereas other control mice show the same result as that of *Prmt5* +/- mice.

Our responses: We thank Reviewer 3 for pointing this out. In an attempt to reduce experimental variability, we have re-examined our flow cytometry gating strategy for dead cell exclusion and CLCA1+ population inclusion and adopted more stringent criteria across all independent experiments in our re-analysis. Results from our re-analysis are now shown in the revised Figure 5C, confirming that the proportions of CLCA1-positive colonic IECs were lowered in the *PRMT5*-deficient mice (**, p<0.01).

This type of comment also applies to the quantification of mucus using WGA staining. The analysis is taken from “a region of interest within one colonic section”. The quantification is performed using the average fluorescent intensity on these colonic sections. These results would also be strengthened by using different types of staining methods or western blot analysis.

Our response: We would like to clarify that WGA staining and Western analysis each address a unique set of questions important for this study. While WGA staining allows for the quantification of overall mucin levels within the epithelial cells as well as the appropriate functions of mucus assembly enzymes involved in building the extracellular mucus structures comprised of multiple different and related mucin proteins in the lumen of the intestine, western blot analysis only reveals the intracellular abundance of the specific protein under investigation. As requested, we included the western results of additional mucus assembly-related proteins from the colonic epithelium whole cell lysates in the revised Supplementary Figure S6A. We confirmed that FCGBP protein

levels in the PRMT5-deficient epithelial cells were significantly lower than those found in wildtype cells. In the same assay, MUC2 and CLCA1 levels were modestly reduced in the PRMT5-deficient cells. It is important to note that these western results do not capture levels of these molecules already secreted to the intestine lumen, which is addressed more appropriately by the WGA assay.

Since the authors have access to organoid cultures and used it to quantify *Clca1* expression, why not use this methodology on *Prmt5* +/- mice and score different parameters to quantify goblet (or tuft) cell maturation and the expression of various effector proteins.

Our responses: We thank Reviewer 3 for this suggestion. However, it remained technically challenging to establish organoid cultures from different donors (e.g. wildtype and mutant mice) with synchronized growth and goblet cell maturation rates. To limit such experimental variability, wildtype organoids cultured in single Matrigel domes were split and replated into two separate Matrigel domes that received the vehicle and inhibitor treatment. In the revised Suppl. Fig. S12B, we included RT-qPCR results for *Fcgbp* and *Muc2*, in addition to *Clca1*, to highlight different goblet cell-related genes that depend on PRMT5 in a methylation-independent and dependent manner. In the revised Discussion (page 10), we discussed the limitation of our findings and areas to be addressed by future studies.

5: Finally, from a mechanistic point of view, *Prmt5* will contribute to the asymmetric methylation of arginine on proteins such as histones. Have the authors attempted to quantify the accessibility of the promoter of candidate target genes such as *Clca1* or *Muc2*? This type of experiment would help demonstrating their point.

Our response: We thank Reviewer 3 for this suggestion. Our response: We thank Reviewer 1 for this suggestion. In an attempt to address these questions, we tested whether it is feasible to employ the Chromatin Immunoprecipitation assay to assess the role of PRMT5 in epigenetic marks deposition on the *Clca1*, *Muc2*, and *Fcgbp* genomic loci. From our ChIP-qPCR experiment using the robust H3K4me3 antibodies (marking active chromatin) on 10 million formaldehyde crosslinked colonic epithelial cells, we were able to observe a robust signal on the housekeeping *Gapdh* gene promoter as expected. However, this assay was unable to pick up signals from the *Clca1* or *Fcgbp* promoters (see figure below). We suspect that this technical challenge is likely due to the limited number of *Clca1* and *Fcgbp*-producing goblet cells present in our sample (less than 10% of the total colonic epithelial cells). We estimated that it will require 30-40 experimental mice per genotype to obtain the sufficient number of purified goblet cells by FACS sorting for this type of assay. In the revised Discussion on page 10, we clarified the current technical challenges for determining whether PRMT5-dependent chromatin modification is involved in its mucosal defense role in the colon.

H3K4me3 enrichment on the *Gapdh*, *Clca1*, and *Fcgbp* promoter from the ChIP-qPCR assay using 10 million colon IECs isolated from WT mice. Each dot represents the result from one mouse. * $P < 0.05$, n.s. not significant (t-test).

In conclusion, this manuscript unravels a putative role of Prmt5 in the control of expression of functional gene programs within specific IEC. While the effects of Prmt5 haploinsufficiency are modest in their experimental setting, these results are of interest but would need to be further supported. Organoids from Prmt5 +/- mice could be better exploited for their demonstration.

Our response: We thank Reviewer 3 for finding our results to be of interest. In the revised Discussion (page 10), we discussed the limitation of organoid experiments and areas to be addressed by future studies.

August 22, 2023

RE: Life Science Alliance Manuscript #LSA-2023-02026R

Dr. Wendy Jia Men Huang
University of California, San Diego
9500 Gilman Drive MC0651
George Palade Laboratories, Room 321B
La Jolla 92093

Dear Dr. Huang,

Thank you for submitting your revised manuscript entitled "The arginine methyltransferase PRMT5 promotes mucosal defense in the intestine". We would be happy to publish your paper in Life Science Alliance pending final revisions necessary to meet our formatting guidelines.

- please add a Running Title and keywords to our system
- please add the Twitter handle of your host institute/organization as well as your own or/and one of the authors in our system
- please add your main, supplementary figure, and table legends to the main manuscript text after the references section
- as there is only one panel in Figures S1, S4, S5, S6, S7, S8, and S13, there is no need to label it as A - please correct figures, their legends, and callouts in the manuscript text accordingly
- please add a call out for Table S1 to your main manuscript text

Figure checks:

- please indicate the scale bar size in the legends for figures 1F,4A, 5D, S5
- Please indicate the molecular weight next to each protein blot

A. FINAL FILES:

B. MANUSCRIPT ORGANIZATION AND FORMATTING:

Sincerely,

Reviewer #1 (Comments to the Authors (Required)):

The addition of data and some discussion greatly improves the manuscript. I gladly recommend the publication of this work.

Reviewer #3 (Comments to the Authors (Required)):

Hernandez JE et al present experimental data supporting a promoting role of the type II arginine methyl transferase gene in mucosal defense in the intestine. Whereas some mechanistic issues could not be solved, the authors have strengthened their results, augmented the number of experimental samples when needed and tested various parameters that altogether confirm their initial results. The manuscript can be published.

August 28, 2023

RE: Life Science Alliance Manuscript #LSA-2023-02026RR

Dr. Wendy Jia Men Huang
University of California, San Diego
9500 Gilman Drive MC0651
George Palade Laboratories, Room 321B
La Jolla 92093

Dear Dr. Huang,

Thank you for submitting your Research Article entitled "The arginine methyltransferase PRMT5 promotes mucosal defense in the intestine". It is a pleasure to let you know that your manuscript is now accepted for publication in Life Science Alliance. Congratulations on this interesting work.

DISTRIBUTION OF MATERIALS:

Again, congratulations on a very nice paper. I hope you found the review process to be constructive and are pleased with how the manuscript was handled editorially. We look forward to future exciting submissions from your lab.

Sincerely,
